# Grafting of Methyl Methacrylate onto Gelatin Initiated by Tri-Butylborane—2,5-Di-Tert-Butyl-*p*-Benzoquinone System

**DOI:** 10.3390/polym14163290

**Published:** 2022-08-12

**Authors:** Yulia Kuznetsova, Ksenya Gushchina, Karina Sustaeva, Alexander Mitin, Marfa Egorikhina, Victoria Chasova, Lyudmila Semenycheva

**Affiliations:** 1Department of Organic Chemistry, Faculty of Chemistry, National Research Lobachevsky State University of Nizhny Novgorod, 23, Gagarin Ave., 603022 Nizhny Novgorod, Russia; 2Federal State Budgetary Educational Institution of Higher Education, Privolzhsky Research Medical University of the Ministry of Health of the Russian Federation, 10/1, Minin and Pozharsky Sq., 603950 Nizhny Novgorod, Russia

**Keywords:** gelatin, poly(methyl methacrylate), tributylborane, radical polymerization, graft copolymer

## Abstract

Graft gelatin and poly(methyl methacrylate) copolymers were synthesized in the presence of the tributylborane—2,5-di-tert-butyl-*p*-benzoquinone (2,5-DTBQ) system. The molecular weight parameters and morphology of the polymer indicate that it has a cross-linked structure. Obtained data confirm the simultaneous formation of a copolymer in two ways: “grafting from” and “grafting to”. It leads to the cross-linked structure of a copolymer. This structure was not obtained for copolymers synthesized in the presence of other initiating systems: azobisisobutyronitrile; tributylborane; azobisisobutyronitrile and tributylborane; azobisisobutyronitrile, tributylborane, and 2,5-di-tert-butyl-*p*-benzoquinone. In these cases, the possibility of the formation of the copolymer, simultaneously in two ways, was excluded. Graft gelatin and poly(methyl methacrylate) copolymers synthesized in the presence of the tributylborane—2,5-di-tert-butyl-*p*-benzoquinone system are promising in terms of their use in scaffold technologies due to the three-dimensional mesh structure, providing a high regenerative potential of materials.

## 1. Introduction

Materials based on biopolymers take a leading position in the development area for the needs of regenerative medicine. They are of interest as a basis for obtaining scaffolds with biomimetic characteristics that supply the necessary conditions for cell growth and the restoration of damaged tissues.

A key advantage of natural polymers such as collagen, fibrin, alginate, gelatin, etc., is their high biocompatibility and biodegradation ability. Collagen and gelatin are widely used as base materials for biomedical needs. Their popularity is due to their natural properties, including weak antigenicity and a high presence in body tissues. It causes their high biocompatibility [1,2,3,4,5,6,7,8]. However, some properties of biopolymers significantly limit their use. For example, biopolymer solutions often have low viscosity, and materials based on them have low mechanical strength. In this regard, hybrid polymers consisting of natural and synthetic ones have been developed to receive materials with improved characteristics compared to natural polymers [1,9,10,11,12,13,14,15,16]. Modeling the process of creating new materials from natural polymers with fragments of synthetic polymers allows the design of the basis of the material to possess combined properties—biological activity from a natural polymer and strength from a synthetic one. This approach makes it possible to obtain the necessary spatial-geometric structure for cell growth. It can provide a high regenerative potential for materials and scaffolds based on hybrid polymers [1,11,12,17,18,19,20,21,22].

The growing demand for new hybrid materials for regenerative medicine is met using various techniques, most often involving the grafting of synthetic fragments onto a natural polymer [1,9,10,11,12,13,14,15,16]. There are different ways to affect both the formation of hybrid polymers and the properties of the resulting materials and scaffolds, including properties that ensure the success of the development of cellular components during cell cultivation or the development of the regenerative process. Such methods include the variation of the synthetic fragment (poly(alkyl (met)acrylates) [23,24,25,26,27,28,29,30], polyurethanes [12], poly-L-lactic acid [11], PVA [31], etc.) and the variety of methods of initiating radical processes, such as radical formations due to thermal decomposition in a defined temperature range [32], photoinitiation [14,15,16,17,26,33,34], or original initiating systems with the inclusion of organoelement compounds [35,36,37,38,39,40].

Thus, the alkylborane–oxygen system initiates the graft polymerization of acrylic monomers onto natural and synthetic polymers by the “grafting from” approach [32,35,36,37,38,39,40]. The growth of the graft polymer begins on macroradicals formed from the polymer due to the separation of the hydrogen atom by initiating radicals. This approach is also possible due to the boration products of collagen or gelatin [32,35,36,37,38,39,40], which contribute to the graft polymerization by the mechanism of reversible inhibition. Initiating systems with trialkylboranes are also attractive because they allow the synthesis of polymer materials in a wide temperature range [32,35,36,37,38,39,40,41,42,43]. In terms of making new initiators, it is interesting that the quinone-type inhibitors of radical processes do not slow down the polymerization of vinyl monomers initiated by systems including trialkylboranes. It was first shown by the polymerization of MMA in the presence of Et_3_B and benzoquinone (BQ) [42,43]. It is well known [44] that quinones form a stable aryloxy radical during the interaction with the macroradical (Figure 1). The formed radical promotes the inhibition of polymerization in the absence of alkylborane. However, if trialkylborane is present in the polymerizate, it interacts with an aryloxy radical. As a result, the polymer chain breaks off with the formation of a boronic fragment at the end of the chain (I and II) and an active alkyl radical due to S_R_2 substitution on the boron atom (Figure 1) [45,46,47].

Using the UV spectroscopy method, it was found [45] that BQ and chloranil mainly form quinoid terminal structures, and 2,5-di-tert-butyl-*p*-quinone and duroquinone form aromatic structures. Both products were found for the remaining *p*-quinones. In addition, chain growth occurs by the mechanism of reversible inhibition due to the labile bond of compound I in the presence of the alkylborane–*p*-quinone system (according to Figure 2). Polymer II does not contain a labile bond for reinitiating polymerization, and it is found mainly in the low-molecular fraction.

Previously, it was established that poly(methyl acrylate) is grafted onto starch by the “grafting to” approach in the presence of the alkylborane–*p*-quinone system due to radical substitution involving phenoxyl macroradicals on the boron atom of borated starch [41].

The implementation of controlled polymerization offers great opportunities for the use of such initiators in macromolecular designs. Previously, statistical copolymers of methyl methacrylate (MMA) and styrene, styrene and tert-butyl acrylate, MMA and butyl acrylate, etc., [48,49,50], block copolymers PS-b-PMMA [51] and gradient copolymers of methyl acrylate and styrene [52], and graft starch and methyl acrylate copolymers [41] were obtained using the alkylborane–*p*-quinone system.

The use of alkylborane as part of the initiator leads to the formation of a graft polymer both by the “grafting from” and “grafting to” method. Therefore, the synthesis of a graft copolymer is possible with a combination of both approaches under certain conditions; that is, it is possible to obtain an interpenetrating polymer network when using a single initiator [53]. In this case, the resulting product must have a cross-linked structure, which is required in scaffold technologies.

In terms of developing new hybrid materials for regenerative medicine based on collagen (gelatin) and acrylic monomers, this work aims to synthesize a copolymer of poly(methyl methacrylate) and gelatin in the presence of a tri-*n*-butylborane (TBB)—2,5-DTBQ system. Therefore, the aims of this study are to obtain information on its molecular weight (MW) characteristics, composition, and structure in comparison with those of similar polymers obtained in the presence of azobisisobutyronitrile (AIBN), TBB, and 2,5-DTBQ; AIBN, TBB; or AIBN, and to identify the prospects for its use in scaffold technologies.

## 2. Materials and Methods

### 2.1. Materials

Organic solvents and methyl methacrylate were purified according to generally accepted methods [54]. Gelatin (Gel) was taken from Dr. Oetker. It contained 87.2% of protein and had values of Mn = 85 kDa and PDI = 1.89.

### 2.2. Synthesis of TBB

Mg chips were taken (19.46 g, 0.8 mol) into a 2 L three-necked flask equipped with a mechanical stirrer and reflux condenser, and the mixture was heated and cooled in an argon atmosphere. Then, BF_3_·Et_2_O (28.2 g, 0.2 mol), iodine crystals, and anhydrous diethyl ether (200 mL) were added to the reaction flask while maintaining the argon atmosphere. The reaction was initiated by a dropwise addition of 9.4 mL of 1-butyl bromide while stirring the reaction mixture, and the remainder of the 1-butyl bromide (73.8 g, 0.6 mol) dissolved in ether (100 mL) was added slowly for 1 h so that the ether refluxed gently. Stirring was continued for an additional 1.5 h following the completion of the addition of 1-butyl bromide, and water (3.6 mL, saturated with ammonium chloride) was added. The reaction raw material was settled, and the clear supernatant ether layer was decanted into the distillation flask. Next, ether was distilled at argon flow, and the residual tributylborane was distilled under vacuum [36]. ^11^B NMR (128 MHz, CDCl_3_) was conducted with a value of δ 86.7.

### 2.3. Polymerization Procedure

Thirty mL of 1% Gel solution was placed in a three-necked flask equipped with a mechanical stirrer and reflux condenser, and it was heated in a water bath to 60 °C in an argon atmosphere. A solution of TBB in pentane was placed in a vacuum ampoule. Pentane was distilled at reduced pressure (10^−2^ Torr), after which the ampoule was filled with argon. The required TBB volume (0.08 g (0.00044 mol)) was taken with an argon-filled syringe and poured into a gelatin solution with intensive stirring. The mixture was kept for 30 min. After that, a degassed solution containing 3 mL of MMA, 0.0154 g (0.25 mol.%) of 2,5-DTBQ, and 0.0046 g (0.1 mol.%) of AIBN was added to the reaction flask. All procedures were carried out in argon flow and constant stirring. The synthesis of the following graft-copolymers was carried out under similar conditions: Gel, TBB, MMA, and AIBN; Gel, MMA, and AIBN; or Gel, MMA, TBB, and 2,5-DTBQ. The residue homopolymer was separated by filtration at the end of copolymerization. A copolymer (Gel, TBB, MMA, and 2,5-DTBQ) with a 1:1 ratio of MMA to gelatin was synthesized using a similar technique, and 0.3 mL of MMA was taken with an argon-filled syringe.

### 2.4. Determination of Unreacted Monomer

Unreacted MMA was measured by bromination according to the Knopp method. Bromine was generated by the reaction of bromide–bromate solution (5.568 g KBrO_3_, 40 g KBr in 1 L of water) with hydrochloric acid. The aqueous dispersion of the copolymer (~2 g and 100 mL of water), 25 mL of bromide–bromate solution, and 10 mL of 10% hydrochloric acid were placed in the flask. The mixture was stirred and left in a dark place for 2.5 h. After that, 15 mL of 10% KI was added, and formed iodine was titrated by 0.1 N Na_2_S_2_O_3_. The blank experiment was conducted with distilled water.

### 2.5. Enzymatic Hydrolysis

The enzymatic hydrolysis of copolymers was carried out using collagenase. The copolymers were previously freeze-dried: Water was distilled under a vacuum under the influence of deep freezing using liquid nitrogen. Collagenase (4% by weight of the copolymer) was added to the lyophilized sample, and then water was added (10 mL per 0.1 g of the copolymer). The mixture was kept for a day, after which the solution was filtered out. The remaining PMMA on the filter was dissolved in chloroform and then concentrated in a round-bottomed flask using a rotary evaporator. The isolated PMMA was analyzed by size-exclusion chromatography (SEC).

### 2.6. Fourier Transform Infrared Spectroscopy (FTIR)

Copolymer films were prepared on KBr plates. The IR absorption spectra were recorded on the “IRPrestige-21” FTIR-spectrophotometer (Shimadzu, Kyoto, Japan).

### 2.7. Size-Exclusion Chromatography

The aqueous dispersion of Gel and PMMA copolymer was analyzed on an LC-20 HPLC system (Shimadzu, Kyoto, Japan) with a low-temperature light-scattering detector ELSD-LT II with the LC-Solutions-GPC software module. Measurements were performed at conditions: The column was Tosoh Bioscience TSKgel G3000_SWxl_ (Tosoh, Tokyo, Japan) with a 5.0 µm pore size, column temperature was equal to 30 °C, the eluent was 0.5 M acetic acid solution with the 0.8 mL/min flow rate, and the injection volume was 20 mL. Narrow disperse dextran standards with a MW range of 1–410 kDa (Fluca) were used for calibration.

Graft PMMA driven out by enzymatic hydrolysis from a copolymer was analyzed on a Prominence LC-20VP system (Shimadzu, Kyoto, Japan) with conditions: Columns were Tosoh Bioscience (polystyrene-divinylbenzene gel, 10^6^ and 10^5^ Å pore size) (Tosoh, Tokyo, Japan), column temperature was equal to 40 °C, and the eluent was THF with a 0.7 mL/min flow rate. A differential refractometer and a UV detector (λ = 254 nm) were used as a detector. Narrow disperse poly(methyl methacrylate) standards were used for calibration.

### 2.8. Surface Morphology Analysis

The surface of gelatin and copolymer samples was studied using a scanning electron microscope JSM-IT300LV (JEOL Ltd., Akishima, Japan) with an electron probe diameter of 4 nm (operating voltage 30 kV) by using detectors of low-energy secondary electrons in a low-vacuum mode to avoid the samples charging. The sponges for the electron microscope were obtained by freeze-drying.

## 3. Results and Discussion

2,5-DTBQ was chosen to study the grafting of MMA onto gelatin in the presence of the alkylborane–*p*-quinone system since the main path of its interaction with the macroradical is carried out by the C = O bond (Figure 1, direction 1). The transition of the quinoid structure to the aromatic one occurs only in this case. On the one hand, it will allow the conduction of copolymerization by the mechanism of reversible inhibition and, on the other hand, it will obtain a colorless product [46]. Copolymer samples were obtained in the presence of AIBN; AIBN and TBB; AIBN, TBB, and 2,5-DTBQ; or TBB. It was necessary to compare the synthesis results in the presence of the listed additives to confirm the mechanism of formation of the final target product.

A two-phase system was obtained during all syntheses, in which the liquid phase was a copolymer solution in water, and the solid phase was a homopolymer. Homopolymer formation was confirmed using FTIR spectroscopy. The IR spectra of the homopolymer isolated from the reaction mixture for all syntheses and the spectra of PMMA obtained by radical polymerization initiated AIBN were identical (Figure 1). The homopolymer yield was determined relative to the initial MMA (Table 1, column 2).

The characteristics of copolymers were studied using physicochemical methods. The IR spectra of copolymers (Figure 2) contain absorption bands related to both gelatin and PMMA. The bands at 3295 cm^−1^ belong to ν(N-H); 2936 and 2916 cm^−1^—ν(C-H); 1634 cm^−1^—ν(COO); ν(N-H); 1537 cm^−1^—δ(N-H) and ν(C-N); and 1238 cm^−1^—ν(C-N). δ(N-H) are distinguished in the spectra of gelatin [55]. In the IR spectra of the MMA, it is a clear absorption band in the area of 1730 cm^−1^ that correspond to the valence vibrations of the carbonyl group. The content of PMMA in the copolymer obtained in the presence of AIBN is insignificant (Figure 2, curve 3), as evidenced by the lowest intensity of the band at 1730 cm^−1^.

Aqueous solutions of freeze-dried samples were treated with the collagenase enzyme to determine the percentage of grafted PMMA in the copolymer. This enzyme destroys gelatin in the copolymer to amino acids [37]. The remaining water-insoluble polymer was weighed to a constant weight after drying. The results of mass loss after enzymatic hydrolysis are shown, in Table 1, as the percentage of grafted PMMA in the copolymer. The injection of TBB into the reaction mass in any combination, as shown in Table 1, significantly increases the yields of both homopolymer and grafted PMMA. Traces of oxygen likely contribute to the formation of a homopolymer due to active alkyl and alkoxyl radicals formed caused by the alkylborane–oxygen system [40]. The increase in the percentage of grafted PMMA in the presence of systems containing tributylborane is due to the feature of such systems to initiate grafting on the surface of various substrates. The highest yields are observed in systems whereby TBB was used with AIBN.

The study of the molecular-mass characteristics of the copolymers showed that MWD curves obtain an additional high-molecular mode (Figure 3, curves 2–4) compared with the MWD of gelatin (Figure 3, curve 1), except for the copolymer synthesized in the absence of AIBN (Figure 3, curve 5). A part of the gelatin likely remains unreacted, and its mode is visible in Figure 3 for all samples. An additional mode should be attributed to the resulting graft Gel-PMMA copolymer. In addition, a low-molecular MWD mode with a molecular weight value of ~10–40 kDa appears in copolymer solutions after synthesis. It is due to the partial hydrolysis of gelatin during synthesis [56]. As a result, the MWD mode corresponding to the initial gelatin shrinks and shifts to the area of smaller molecular weights (Figure 3, curves 2–5). The high-molecular MWD mode corresponding to the graft copolymer (Figure 3, curve 4) is minimal for the sample synthesized with AIBN as an initiator in order to meet the lowest value of the percentage of grafted PMMA. The MWD curve has two modes that are almost equal in height corresponding to gelatin and the polymer grafted to it for the sample synthesized in the presence of AIBN and TBB (see Table 1, line 2). Quite unexpectedly, there is no high–molecular mode on the MWD curve of the copolymer synthesized in the presence of the TBB—2,5-DTBQ system (Figure 3, curve 5), although the percentage of grafted PMMA in the copolymer is 49% (see Table 1, line 5). We believe that the graft copolymer has a cross-linked structure in this case, which does not allow for analyzing a sample by SEC without pre-hydrolysis by enzymes. It is worth noting that all copolymers obtained in the presence of TBB have a significant increase in the low-molecular mode on the MWD curves. However, the fraction corresponding to this mode does not degrade even under the action of enzymes [37]. Therefore, in reality, it is a decrease in the intensity of the mode that corresponds to the copolymer.

PMMA isolated from copolymers by enzymatic destruction of the protein in its composition was studied to clarify the assumption about the cross-linked nature of the copolymer synthesized in the presence of the TBB—2,5-DTBQ system. The SEC method detected a high-molecular PMMA with a unimodal MWD curve in all cases (Figure 4). Differences in MW values can be associated with different polymerization mechanisms.

There is a high probability that the first stage of formation of the graft copolymer in all cases is the boration of gelatin, according to Figure 5(1). Furthermore, PMMA macroradicals formed due to active alkyl and alkoxyl radicals during the oxidation of TBB, and AIBN can interact with borated gelatin in the case of using the TBB—AIBN system (Figure 5(2a)) and implement the “grafting to” approach.

The 2,5-DTBQ accepts the PMMA macroradical in the case of systems with it. Stable aryloxy radicals enter into S_R_2 substitution on the boron atom of borated gelatin and implement the “grafting from” approach (Figure 5(2b)). We think that embedded aromatic fragments can initiate polymerization by the mechanism of reversible inhibition.

The formation of a copolymer according to Figure 5(2a) is not excluded in the system initiated by AIBN, TBB, and 2,5-DTBQ. Therefore, chain transmission can pass due to both the alkylborane and the alkylborane–*p*-quinone system. It leads to the formation of a copolymer with a synthetic part with a wider MWD curve (Figure 4, curve 1). When the TBB—2,5-DTBQ system initiates copolymerization, macroradicals are formed mainly due to the reversible dissociation of borated gelatin [35,36,37] (Figure 6 (in a blue frame)). We cannot exclude radical formation due to the oxidation of TBB [41]. Apparently, the formed macroradicals react with *p*-quinone by a radical-substitution reaction on the boron atom and form a cross-linked polymer (Figure 6). Thus, the growth of the graft chain begins (“grafting from”) and ends (“grafting to”) on gelatin. The percentages of grafted PMMA (Table 1) and MW (Figure 4, curve 3) are lower than in the other samples. This is due to the rate of pseudo-living polymerization being significantly less than the rate of traditional polymerization in the presence of AIBN. The unreacted *p*-quinone after polymerization in the presence of the TBB—2,5-DTBQ system was detected by UV-Vis in contrast to using the initiating system AIBN, TBB, and 2,5-DTBQ. It indicates the complexity of *p*-quinone embedding into the polymer chain in the absence of a radical initiator. Thus, it likely interacts with the macroradical at the time of the “revival” of the polymer chain.

PMMA isolated from copolymers synthesized in the presence of 2,5-DTBQ by enzymatic hydrolysis was analyzed using SEC with a refractometric and UV detector (λ = 254 nm). The area of molecular weights on the MWD registered on both detectors coincides (Figure 7). Therefore, all polymer molecules contain an embedded *p*-quinone, which is a confirmation of the proposed mechanism.

In the previously shown examples of the synthesis of the graft Gel-PMMA copolymer, the reactions proceeded at gelatin to MMA with a mass ratio of 1:10. As already noted, a significant amount of homopolymer formed along with the copolymer in all cases (Table 1). It is an avoided impurity in the final product if its use is focused in scaffold technologies. In addition, a large amount of unreacted MMA remains after synthesis. It is toxic, which also complicates the technology of obtaining the target product since additional purification is required. Therefore, the concentration of MMA was reduced for further studies in order to only obtain the graft copolymer. The copolymer with a Gel to MMA mass ratio of 1:1 was synthesized in the presence of TBB and 2,5-DTBQ under the same conditions as the previous samples. Neither PMMA homopolymer nor unreacted MMA was found in the chloroform extract from a reaction mixture after the end of the process. This indicates that the entire initial monomer is part of the copolymer. The MWD curve of the copolymer solution (Figure 8, curve 2) is unimodal, similarly the the case of obtaining a copolymer from a reaction mixture with a high MMA content (Figure 3, curve 5). It is probable that the formed copolymer has a cross-linked structure, and SEC does not register it, as in the previous example.

SEM showed that the copolymers obtained using different initiating systems differ (Figure 9). The SEM image of the copolymer synthesized in the presence of AIBN (Figure 9b) is close to the original gelatin (Figure 9a), which correlates with the lowest percentage of grafted PMMA. Copolymers synthesized using AIBN and TBB (Figure 9c) and AIBN, TBB, and 2,5-DTBQ (Figure 9d) have equal surfaces, which correspond to similar synthesis parameters, copolymer characteristics, and proposed mechanisms. Likely, the “grafting to” approach is implemented in both copolymers, but pseudo-living polymerization is added to the system with *p*-quinone. The structure of the copolymer synthesized in the presence of the TBB—2,5-DTBQ system is a cellular frame (Figure 9e). It confirms the formation of a cross-linked copolymer due to the implementation of both approaches “grafting to” and “grafting from”, which is very promising for its use in scaffold technologies. The morphology of the copolymer synthesized with a Gel to MMA mass ratio of 1:1 (Figure 9f) confirms this assumption, which is similar to the morphology of the copolymer obtained in excesses of MMA (Figure 9e), although the pore size, in this case, is much larger. This is due to the low concentration of MMA in the reaction mixture.

## 4. Conclusions

Thus, a graft copolymer of PMMA onto gelatin was obtained and characterized in the presence of tributylborane and 2,5-di-tert-butyl-*p*-benzoquinone at different mass ratios of gelatin to MMA. It was found that all MMA passes into the grafted PMMA with a mass ratio of Gel:MMA = 1:1. The information on the composition, molecular weight parameters, and morphology indicates that the graft Gel-PMMA copolymer, in this case, has a cross-linked structure and thus is very promising for its use in scaffold technologies. The results of studying the properties (molecular weight characteristics and surfaces) of graft Gel-PMMA copolymers synthesized in the presence of different initiating systems (TBB and 2,5-DTBQ; AIBN, TBB, and 2,5-DTBQ; AIBN and TBB) allowed establishing that the proportion of grafted poly(methyl methacrylate) at the ratio Gel:MMA = 1:10, as well as the homopolymer yield, depends on the composition of the initiator, and its values are greater when tri-*n*-butylborane is a part of the initiating system. In addition, the conducted studies allowed reasoning that the graft polymerization of MMA onto gelatin in the presence of the alkyl borane–*p*-quinone system combines two approaches: “grafting to” and “grafting from”. Notably, the analysis of the morphology of sponges based on graft-copolymers obtained by freeze-drying showed the presence of a structure with a system of interconnected heterogeneous pores. The latter testifies in favor of the possibility of creating promising materials and scaffolds for regenerative medicine with structural characteristics similar to sponges, which will allow cells to be placed in the material and ensure the exchange of gases, liquids, and cell waste products.

## Data Availability

Not applicable.

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
