# Peer review of "Grafting of Methyl Methacrylate onto Gelatin Initiated by Tri-Butylborane—2,5-Di-Tert-Butyl-p-Benzoquinone System"

_polymers, 2022, doi:10.3390/polym14163290_

Round 1
Reviewer 1 Report
Dear Authors
The presented data in your manuscript is very interesting for the readers and treats an essential point of interest in the development of polymer scaffolds for tissue regeneration.
The design of the research is well organized and the presentation and discussion of the obtained data are clear enough to declare the manuscript idea.
The novelty of the research idea is clear enough.
The only comment I have on this point is the similarity of the keys in the figures which makes it difficult to follow and definite between the different synthetic copolymers using different initiation systems. Please change it to be easier to recognize and increase the resolution of the figures.
Only, a minor revision is required to consider your manuscript for publication.
Author Response
Dear Reviewer,
Thank you for your review of our paper and its appreciation! We have corrected the figure according to your comment (The only comment I have on this point is the similarity of the keys in the figures which makes it difficult to follow and definite between the different synthetic copolymers using different initiation systems. Please change it to be easier to recognize and increase the resolution of the figures.).
Sincerely,
Authors
Reviewer 2 Report
The current manuscript reports on new grafting technique on gelatin using tributylborane-2, 5-di-tert-butyl-p-benzoquinone system especially for regenerative medicine applications. The technique proposed in this manuscript, grafting from and grafting to at the same time, is interesting and promising as a scaffold preparation method. In addition, because the gelatin, initiators, and monomers used in this manuscript are very common, a lot of readers would be interested. I just noticed a slight redundancy in logic. Therefore, I recommend minor revision.
Below are the comments to the manuscript:
1) For the readability of the manuscript, the abbreviation 2, 5-DTBQ should be appeared in the abstract.
2) Is there an approache utilizing an interpenetrating polymer network (IPN) for the preparation of scaffold for regenerative medicine ? If YES, the research should be cited in the Introduction section.
3) It is easy to read, but I felt there were a few too many paragraphs.
4) To keep the logic nice and simple, I felt that the results and discussion related to Figure 9 should be before Figure 8.
Author Response
Dear Reviewer,
Thank you for your review of our paper and its appreciation! Our answers to your points are as follows.
1) For the readability of the manuscript, the abbreviation 2, 5-DTBQ should be appeared in the abstract.
Response: Agree with the remark. The corresponding changes have been made to the text.
2) Is there an approache utilizing an interpenetrating polymer network (IPN) for the preparation of scaffold for regenerative medicine? If YES, the research should be cited in the Introduction section.
Response: Thank you for your comments; we have made an appropriate clarification in the introduction.
3) It is easy to read, but I felt there were a few too many paragraphs.
Response: The number of paragraphs was reduced in accordance with the remark.
4) To keep the logic nice and simple, I felt that the results and discussion related to Figure 9 should be before Figure 8.
Response: The corresponding changes have been made to the text.
Sincerely,
Authors